# Emerging Prognostic and Predictive Significance of Stress Keratin 17 in HPV-Associated and Non HPV-Associated Human Cancers: A Scoping Review

**DOI:** 10.3390/v15122320

**Published:** 2023-11-25

**Authors:** Taja Lozar, Wei Wang, Niki Gavrielatou, Leslie Christensen, Paul F. Lambert, Paul M. Harari, David L. Rimm, Barbara Burtness, Cvetka Grasic Kuhar, Evie H. Carchman

**Affiliations:** 1McArdle Laboratory for Cancer Research, University of Wisconsin School of Medicine and Public Health, Madison, WI 53705, USA; tlozar@wisc.edu (T.L.);; 2University of Wisconsin Carbone Cancer Center, Madison, WI 53705, USA; 3University of Ljubljana, 1000 Ljubljana, Slovenia; 4Department of Pathology, Yale University, New Haven, CT 06510, USA; 5Ebling Library, University of Wisconsin School of Medicine and Public Health, Madison, WI 53705, USA; leslie.christensen@wisc.edu; 6Department of Human Oncology, University of Wisconsin School of Medicine and Public Health, Madison, WI 53705, USA; 7Department of Medicine and Yale Cancer Center, Yale School of Medicine, New Haven, CT 06510, USA; 8Institute of Oncology Ljubljana, 1000 Ljubljana, Slovenia; 9Department of Surgery, University of Wisconsin School of Medicine and Public Health, Madison, WI 53705, USA; 10William S. Middleton Memorial Veterans Hospital, 2500 Overlook Terrace, Madison, WI 53705, USA

**Keywords:** prognostic biomarker, predictive biomarker, cytokeratin 17, stress keratin 17

## Abstract

A growing body of literature suggests that the expression of cytokeratin 17 (K17) correlates with inferior clinical outcomes across various cancer types. In this scoping review, we aimed to review and map the available clinical evidence of the prognostic and predictive value of K17 in human cancers. PubMed, Web of Science, Embase (via Scopus), Cochrane Central Register of Controlled Trials, and Google Scholar were searched for studies of K17 expression in human cancers. Eligible studies were peer-reviewed, published in English, presented original data, and directly evaluated the association between K17 and clinical outcomes in human cancers. Of the 1705 studies identified in our search, 58 studies met criteria for inclusion. Studies assessed the prognostic significance (n = 54), predictive significance (n = 2), or both the prognostic and predictive significance (n = 2). Altogether, 11 studies (19.0%) investigated the clinical relevance of K17 in cancers with a known etiologic association to HPV; of those, 8 (13.8%) were focused on head and neck squamous cell carcinoma (HNSCC), and 3 (5.1%) were focused on cervical squamous cell carcinoma (SCC). To date, HNSCC, as well as triple-negative breast cancer (TNBC) and pancreatic cancer, were the most frequently studied cancer types. K17 had prognostic significance in 16/17 investigated cancer types and 43/56 studies. Our analysis suggests that K17 is a negative prognostic factor in the majority of studied cancer types, including HPV-associated types such as HNSCC and cervical cancer (13/17), and a positive prognostic factor in 2/17 studied cancer types (urothelial carcinoma of the upper urinary tract and breast cancer). In three out of four predictive studies, K17 was a negative predictive factor for chemotherapy and immune checkpoint blockade therapy response.

## 1. Introduction

Advances in our understanding of the molecular biology of cancer and the development of novel treatment modalities have led to the improved survival of cancer patients [1]. The ability to characterize various tumor types at the genome, transcriptome, and proteome levels has changed our approach to treatment in modern-day oncology from a one-size-fits-all approach to highly personalized precision medicine using targeted therapeutics [2]. One of the key challenges in the era of precision medicine remains the identification and validation of robust prognostic and predictive biomarkers, which can (1) help select patients who are more likely to experience a more aggressive disease course, which requires more aggressive treatment or additional therapies; and (2) identify the most appropriate treatment with which to minimize treatment-related adverse events and maximize clinical benefit. Prognostic biomarkers aim to achieve the first goal and inform a likely cancer outcome (such as disease recurrence, progression, or death) irrespective of treatment. Predictive biomarkers, on the other hand, are treatment selection markers and are, by definition, treatment-dependent. They inform a likely effect of a specific therapeutic intervention in a patient, aiming to achieve the second goal [3].

Biomarker development, similar to therapeutic drug development, is driven by the discovery of genes or proteins that are proven to have biological relevance in the development of cancer [4]. One such protein is stress keratin (or cytokeratin) 17 (K17), which belongs to a group of acidic (type I) keratins [5]. K17 is a type of intermediate filament protein that is normally expressed during embryogenesis, induced in response to tissue injury, silenced in mature somatic tissues except in certain stem cell populations [6,7], and re-expressed in some cancers [8,9]. The expression of K17 is characterized as high/medium in colorectal, head and neck, stomach, pancreatic, urothelial, breast, cervical, skin, ovarian, lung, endometrial, carcinoid, and thyroid cancers [5]. The exact mechanism by which K17 is driving the prognosis of these cancers remains to be defined. K17 appears to be involved in multiple carcinogenesis pathways, such as transcription regulation and subcellular localization [10,11], glycolysis [12], enhancing cancer stemness [13], immune evasion [14,15], and others [5,16]. Recent preclinical data have suggested that viral oncogenes, such as the mouse papillomavirus mMuPV1 E6 and E7, can induce the expression of K17 in both cutaneous and mucosal sites [14,17] and thereby suppress host immune response [14]. Specifically, the virus appears to induce the expression of K17, which, in turn, suppresses T cell-mediated immune surveillance by suppressing the macrophage-mediated C-X-C Motif Chemokine Ligand 9/10 (CXCL9/CXCL10) chemokine signaling involved in attracting activated CD8+ T cells into tissues, resulting in decreased CD8+ T cell infiltration. Similar observations have been reported in studies in cutaneous basal cell carcinoma, showing that K17 levels directly correlate with changes in the expression of inflammatory T-helper cytokines [8,18], and colocalization of K17 with key cytokines, including CXCR3, CXCL10, and CXCL11 [19], as well as in a cervical cancer mouse model where K17 high expressing lesions had increased transcript levels of pro-inflammatory cytokines, including interferon gamma, CXCL9, CXCL10 and CXCL11 [20]. Recent data suggest that K17 supports viral persistence as well as disease severity in the female reproductive tract and modulates the tumor immune microenvironment in papillomavirus-associated lesions [17]. A comprehensive review of the role of K17 in the hallmarks of cancer has recently been published elsewhere [5].

While studies targeting K17 for anticancer drug development remain limited [21], a growing body of literature suggests that the expression of K17 correlates with clinical outcomes across various cancer types, establishing its role as a prognostic biomarker [22,23,24,25,26,27,28,29,30,31]. In addition, recent data published by our group and others suggest an emerging predictive role of K17 in selected cancer types for both chemotherapy and immunotherapy response [15,31]. In this scoping review, we present and map the available clinical evidence of the prognostic and predictive value of K17 in human cancers, particularly those of viral origin, identify potential knowledge gaps, and discuss emerging concepts and clinical opportunities for K17 in the era of immuno-oncology. Considering the preclinical data suggesting that an infection with an oncogenic virus such as human papillomavirus (HPV) can induce K17 expression, we review these data in relation to HPV-associated vs. non-associated cancers.

## 2. Materials and Methods

### 2.1. Method Selection

Due to the heterogeneity of investigated cancer types, K17 assessment and scoring methods, patient demographics, and effect size variables reported by the studies, a scoping review of the literature was selected as the most appropriate methodology for evidence synthesis [32] and was reported following the Preferred Reporting Items for Systematic Reviews Extension for Scoping Reviews (PRISMA-ScR) guidelines [33]. The Population/Concept/Context (PCC) question was “What is the prognostic and predictive significance of stress keratin 17 expression in human cancer patients?” A comprehensive search of PubMed, Web of Science, and Scopus was conducted (June 2023). No language, article type, or publication date limits were placed on the search.

### 2.2. Eligibility

Publications meeting the following inclusion criteria were included in this review: peer-reviewed papers in English that presented original data investigating the prognostic or predictive role of K17 in human cancers. Only studies involving human participants were included. Eligible studies evaluated K17 status at the RNA or protein level along with a measure of the clinical outcome and directly addressed the association between K17 status and the clinical outcome. Eligible studies adequately defined the methods used, such as (a) the method of RNA data analysis for transcriptome studies; or (b) the antibody, marker cut-off, and scoring system used for IHC studies. Studies featuring K17 as part of a combination of prognostic markers or a prognostic gene signature were excluded. A list of studies including CK17 as a part of a prognostic gene/protein signature without available prognostic assessment as a stand-alone marker is available in Appendix A. Other exclusion criteria included review articles, books, book chapters, and studies using preclinical data.

### 2.3. Information Sources and Searches

The review team collaborated with a research librarian (LC) to develop and execute a comprehensive search of the literature. This search used controlled vocabulary and keywords related to K17 expression in human cancers. Appendix A lists full electronic search strategies. A search was designed in PubMed and translated into the following databases: Embase via Scopus (Elsevier Science, Amsterdam, The Netherlands), Cochrane Central Register of Controlled Trials (CENTRAL) via Cochrane Library (Wiley, New York, USA), Web of Science Core Collection (Clarivate), and Web of Science Preprint Citation Index (Clarivate, Philadelphia, USA). All searches were conducted from database inception to 5 July 2023. Animal studies were removed by applying an exclusion filter in PubMed and Scopus. No language, age, or other restrictions were applied to the search results. A Google Scholar search was executed on 5 July 2023, and the first 200 results, sorted by relevance, were exported. The results were downloaded into EndNote (Clarivate) and underwent manual deduplication by the research librarian. Unique records were uploaded to Covidence (Veritas Health Information, Melbourne, Australia) for screening and review by the study team using pre-determined inclusion/exclusion criteria. 

### 2.4. Study Selection

The titles and abstracts were independently screened by two investigators (T.L., E.C.). Disagreements were resolved by a third reviewer (C.G.K.). Full-text screening was performed by two or three reviewers (T.L., E.C., W.W.). Disagreements were resolved by consensus.

### 2.5. Data Collection and Synthesis of Results

A data-charting form was developed by T.L. to determine which variables to extract. Initially, two reviewers (T.L. and E.C.) began charting 10% of the studies, discussed the results, and updated the data-charting form. Data extraction for all studies was performed in duplicate (by two of the following investigators: T.L., E.C., W.W., C.G.K.). Any disagreements were resolved through discussion between the reviewers. Data from eligible studies were extracted using a standardized form. The form captured the relevant information on key study characteristics (first author, year, country, cancer type, study design, tissue type) and information on whether the studies reported an effect size of the marker on clinical outcome and the value. Studies were grouped by the K17 evaluation method (RNA/protein/other). For transcriptome studies, the source of the data and analysis method used, along with the effect size (if reported) or key findings (positive/negative association between marker and outcome, *p* value), were extracted. For proteomics studies, additional data on antibody used, scoring methodology, and marker cut-off were collected along with effect size and key findings.

### 2.6. Quality Assessment and Risk of Bias

Critical appraisal of the included studies is not required for scoping reviews [32,33] and was not performed in this study (with potential impact elaborated in the Discussion section).

## 3. Results

### 3.1. Literature Search

The search strategy identified 1705 unique studies. Based on title and abstract screening, 1503 references were excluded. Of the remaining 202 studies, 144 were excluded based on the full text. The remaining 58 studies were included in this scoping review (Figure 1). 

### 3.2. Included Studies’ Characteristics

The characteristics of eligible studies are presented in Table 1 and Figure 2. Briefly, eligible studies were published between 2004 and 2023 and were predominantly conducted in China (20/58, 34.5%), followed by the USA (14/58, 24.1%) (Figure 2B,C). Fifty-four (93%) assessed solely the prognostic significance of K17; two studies (3.4%) assessed the predictive significance of K17; and two studies (3.4%) assessed both the prognostic and predictive significance. Altogether, 11 studies (19.0%) investigated the clinical relevance of K17 in cancers with a known etiologic association to HPV; of those, 8 (13.8%) were focused on head and neck squamous cell carcinoma (HNSCC) and 3 (5.1%) on cervical squamous cell carcinoma (SCC).

Of the included studies, 32.8% evaluated K17 expression at the RNA level, 55.2% at the protein level, and 12.1% performed both RNA and protein assessment for prognostic and/or predictive value analysis. One study (1.7%) was performed on serum samples [34], and one study (1.7%) was performed on cytology samples [35]. The remaining studies were performed on histology samples. Studies were analyzed separately based on the clinical significance being evaluated (prognostic, predictive) and based on a K17 evaluation method (RNA, protein).

#### 3.2.1. Prognostic Significance

Since 2004, 56 studies investigated the prognostic significance of K17 in human cancers, and of those, 32 (57.1%) studies assessed K17 expression at the protein level, 18 (32.1%) at the RNA level, and 6 (8.9%) at the RNA and protein level (Table 2, Figure 3). The first published studies on the prognostic significance of K17 in human cancers were performed in urothelial [36] (2004) and breast carcinoma [37] (2007) using immunohistochemistry (IHC). Initial studies evaluating K17 at the transcriptome level were performed in oral HNSCC [38] (2017) and TNBC [29] (2017). To date, HNSCC [15,24,38,39,40,41,42], TNBC [28,29,43,44,45,46,47], and pancreatic cancer [31,35,48,49,50,51] are the most studied cancer types (n = 7). K17 was a significant prognostic variable in 16/17 (94.1%) investigated cancer types and 43/56 (76.8%) studies (Table 2). 

**In HPV-associated cancers**, seven (12.5%) studies investigated the prognostic role of K17 in HNSCC [24,38,39,40,41,42,52] and three (5.4%) studies in cervical SCC [22,34,53]. One study (1.8%) in HNSCC investigated both the prognostic and predictive role [15]. In HNSCC, RNA-based studies have consistently found an inverse correlation between K17 expression and clinical outcome, whereas protein-based studies reported varying results using different scoring methodologies. Two studies investigated the correlation between K17 protein expression and prognosis in oral HNSCC and did not find a significant correlation [40,41], whereas an RNA-based study in oral HNSCC found a negative correlation [38]. Our group, as well as Regenbogen et al., found a significant association in HNSCC using a protein-based K17 assessment, particularly in oropharyngeal SCC [15,24]. Regenbogen further found that K17 expression predicted survival, whereas HPV status did not, and that high K17 expression status, when combined with negative HPV status, was a significant predictor of poorer survival [24]. In cervical SCC, two studies investigated the correlation between K17 protein expression and clinical outcome. Escobar-Hoyos used a standardized scoring approach used by their group across various solid tumors and found a significant negative correlation [22], whereas Hashiguchi et al. did not [53]. He et al. quantified the circulating K17 in the serum of cervical cancer patients regardless of the histologic subtype and found that high K17 was prognostic of overall survival [34].

Studies investigating **K17 protein expression** (n = 38; 36 protein only and 2 RNA and protein) have found K17 to hold negative prognostic value in the vast majority of cancer type (breast [37], cervical [19,31,50], colorectal [54,55], endometrial [30], esophageal [56,57], gallbladder [58,59], gastric [26,60,61], HNSCC [15,24], ovarian cancer [27], pancreatic [31,35] and renal carcinoma [62], and TNBC [43,63]). Evidence of positive prognostic value of K17 protein expression has been reported in some studies in bladder [64], HER2+ breast [65], colorectal [66], HNSCC [39], and urothelial carcinoma [36]. Individual studies in bladder [67], cervical [53], ovarian [68], HNSCC [40], and TNBC [29,44,45,46,47] have found no correlation between K17 protein expression and outcome. 

The detailed characteristics and key findings of prognostic protein expression studies are presented in Table 3 and Appendix A. Study cohorts ranged from 26 to 692 patients (Table 3). The percentage of K17 high expressors varied by cancer type and ranged from 12% to 95% (Table 2). Thirty studies (78.9%) reported effect size. IHC was the most commonly used method to determine K17 expression (36/38, 94.7%) 22. 11. 2023 08:34:00. One study (2.6%) used immunofluorescent staining to determine K17 expression in HNSCC [15], and another study (2.6%) used an ELISA assay with spectrometry to quantify circulating K17 in the serum of patients with cervical cancer [34]. Eighteen studies [22,26,29,36,37,39,41,44,47,55,56,58,59,60,62,63,64,68] (32.1%) used tissue microarrays (TMAs) for biomarker evaluation. One study (2.6%) used an automated software to score K17 expression [15]. The remaining studies performed manual assessment by pathologists using different semiquantitative scoring approaches. Several studies performed semiquantitative assessment using PATHSQ (which scores the percentage of 2+ strong positive tumor cells) across various cancer types, such as pancreatic, esophageal, cervical, endometrial, HNSCC, and colon cancer [22,23,24,30,35,48,54,57]. Studies in renal cell carcinoma, TNBC, and breast cancer have evaluated only the presence or absence of any stained cells [37,43,62,69]. Twelve studies (31.6%) determined the biomarker cut-off experimentally [22,23,24,30,31,35,44,48,55,59,62,64]. One study in colorectal cancer (2.6%) included an interobserver variability assessment for the K17 assay [55]. The details of biomarker assessment methodologies are presented in Appendix A.

Studies investigating **K17 at the transcriptome level** (n = 24) have found K17 to hold negative prognostic value in most cancer entities (endometrial [30,71], esophageal [57], gastric [72], HNSCC [38,42,52], lung cancer [73,74], melanoma [75,76], pancreatic adenocarcinoma [50,51], and renal cell carcinoma [77]). In TNBC, a negative correlation between K17 RNA expression and outcome was found only in a subgroup of patients with invasive ductal carcinoma with large tumors and advanced stage [29]. A positive correlation between K17 and clinical outcome was found in breast [65,78] and gastric cancer [77]. Studies in pancreatic cancer [49] and primary central nervous system (CNS) lymphoma [79] found no correlation.

Two pan-cancer analyses have found a negative correlation in renal, hepatocellular, lung and pancreatic adenocarcinoma, melanoma, mesothelioma, and endometrial and bladder urothelial carcinoma (not included in Table 2) [71,80]. Another pan-cancer analysis found a positive correlation between breast, thyroid renal, and endometrial cancer [81].

The detailed characteristics and key findings of prognostic transcriptome-based studies are presented in Table 4 and Appendix A. Thirteen studies (56.5%) were based on data derived from The Cancer Genome Atlas (TCGA) [30,31,49,50,57,71,71,73,75,76,80]. Five studies (21.7%) generated their own transcriptome data [38,42,51,52,79]. Thirteen studies reported an effect size in their findings [30,31,42,50,57,65,71,72,75,78,79,80,81]. 

#### 3.2.2. Predictive Significance

Of the 58 studies included in this review, 4 studies (6.9%) investigated the predictive significance of K17 in breast [69], colorectal [66], HNSCC [15], and pancreatic adenocarcinoma [83]. Two studies were based on chemotherapy-treated cohorts [69,83], and two studies investigated immunotherapy-treated cohorts [15,66]. Effect size was reported by two studies [69,83]. A study by Diallo et al. performed K17 IHC assessment on a TMA in a prospective cohort of 224 breast cancer patients treated with high-dose vs. Dose-dense chemotherapy regimens and found that K17 was a negative predictive factor of OS for patients treated with the dose-dense but not high-dose regimen [69]. The Shroyer lab reported that K17 expression is predictive of shorter overall survival (OS) in gemcitabine-treated pancreatic ductal adenocarcinoma based on transcriptome data [83]. Our group found that K17 expression by IHC was predictive of an inferior response to pembrolizumab in a pilot cohort of 26 HNSCC patients regardless of HPV status [15]. Conversely, Liang et al. found that K17 by IHC was predictive of a superior response to anti-PD1 blockade in a pilot cohort of 30 missmatch repair deficient (dMMR) colorectal cancer patients, indicating a positive correlation in this cancer type [66]. This group performed an ROC analysis and found an AUC of 95% for high K17 to predict response. The characteristics of predictive studies are presented in Table 5 and Appendix A. 

## 4. Discussion

In this scoping review, we sought to identify the available evidence of the prognostic and predictive value of K17 in human cancers and to examine how research has been conducted in this field, with an emphasis on HPV-associated cancers. The majority of the studies on the clinical significance of K17 in human cancers were performed in cancers of non-viral origin. We do, however, believe that the present manuscript addresses an important knowledge gap with respect to how research into the clinical relevance of K17 is currently conducted and to which cancer types may be of interest for future studies. Our findings suggest that head and neck cancer (a cancer type with a known association to HPV infection relative to anatomic location) is among the cancer types with the strongest evidence for clinical translation. In reviewing the methodology used, we found that most studies used IHC to evaluate K17 expression. We found that prognostic studies in the retrospective setting were the most frequently conducted studies and that breast cancer, particularly TNBC, pancreatic cancer, and HNSCC, are the most frequently investigated cancer types. In most cancer types, including HPV-associated cervical cancer and HNSCC, high expression of K17 at the RNA and/or protein level showed negative prognostic impact. Positive prognostic value was shown in urothelial carcinoma of the upper urinary tract and in breast cancer. In bladder and ovarian cancer, conflicting results and unvalidated assays hinder our ability to draw meaningful conclusions. We further found that prognostic value of K17 differs between cancer types and even within a specific cancer type (such as breast cancer). Our review also revealed negative predictive value in three of four studies that have investigated the predictive significance of K17 in chemotherapy-treated breast and pancreatic cancer and immune checkpoint blockade-treated HNSCC and colorectal cancer. 

Evidence consistently suggests that K17 is a negative prognostic biomarker in melanoma, endometrial, esophageal, gallbladder, lung, and renal carcinoma. Conflicting data exist with respect to cervical cancer, HNSCC, colorectal, gastric, and pancreatic cancer; however, over half of the published studies have found K17 to hold negative prognostic value in these cancer types. Similarly, the majority of evidence suggests that K17 is a positive prognostic biomarker in TNBC and invasive breast cancer and in transitional cell urothelial carcinoma of the upper urinary tract. On the other hand, no evidence for prognostic significance of K17 expression was found in primary CNS lymphoma. The prognostic value of K17 is therefore cancer-type-dependent.

Our descriptive analysis of the conclusions made by studies evaluating K17 expression at the protein vs. RNA level showed that the direction of association between K17 and clinical outcome appears consistent in breast, endometrial, esophageal, and renal cancer regardless of the assessment method used. We noted discrepancies particularly in cancer types that have been more extensively studied in the recent years, such as HNSCC and pancreatic cancer. The discrepancies between RNA and protein expression in human tissue have been appreciated for some time [84]. mRNA levels primarily determine the protein amounts; however, due to the delay between transcription and translation, RNA and protein levels are not necessarily synchronized at the single-cell level [85]. Some of the key factors affecting this relationship are the abundance of the protein in question (for high abundance proteins, the production rate tends to saturate due to the saturation of ribosomes on mRNAs, leading to high abundance proteins often having the lowest protein turnover rates [86,87,88,89]), as well as the compartmental correlation in the cell due to the localizaton of both molecules [85]. To adequately address these discrepancies, one would need to quantify RNA and protein abundance and kinetics in different subcellular components at both steady state and during dynamic transitions, which is limited by the capabilities of current technologies [85]. Since no such data exist for K17 at this time, we can only assume that these dynamics may be the underlying cause of some of the discrepancies between observed findings based on RNA vs. protein data, and possibly within a specific cancer type. 

As pointed out by Kerr et al. in a recent review on prognostic biomarkers for precision medicine, clinically useful biomarkers should focus on a specific tumor stage, be clinically actionable, and reliably estimate an effect (sufficiently powered) [90]. The authors further discuss an arbitrary cut-off for hazard ratios to be set at 2. With this arbitrary cut-off in mind, studies in cervical adenocarcinoma [23], HNSCC [24,41], colorectal [55], esophageal [56,57], gallblader [58,59], and pancreatic [31,83] carcinoma suggest a high clinical utility for K17 as a negative prognostic biomarker.

Biomarker development has historically been driven by the discovery of genes involved in the regulation of key carcinogenic processes in specific tumor types (such as estrogen receptor in breast cancer, BRC-ABL in chronic myeloid leukemia, etc.) and has been tightly linked to cancer drug development. What makes K17 particularly interesting are the observations that it can be induced in response to viral infection. A study in a mouse papillomavirus infection model revealed that K17 can be induced by the viral oncoproteins E6 and E7 in the skin [14] and cervicovaginal tract [17]. This led to studies of the prognostic significance of K17 in the context of HPV-related cancers, such as HNSCC and cervical squamous cell carcinoma. In the cervix, several studies have demonstrated negative prognostic value [22,53]. In HNSCC, K17 alone or in combination with HPV status can predict shorter os (K17 high, HPV-negative) [24]. A recently published study (not included in this review) showed that K17 status is independent of HPV status [91] and that K17 holds prognostic significance in both HPV-positive and -negative HNSCC [15]. In addition, recent studies, including preclinical studies in HPV infection models, have found that K17 may be contributing to changes in tumor T cell inflitration [14,15,66], which is a critical mechanism of immune escape [92]. This led to the study of the role of K17 in mediating the tumor immune response and its role as a potential biomarker of the response to immune checkpoint inhibition. Recent studies in HNSCC and colorectal cancer have revealed opposite roles for K17 in immune modulation. In HNSCC patients and the MOC2 syngeneic mouse model that is considered immunologically “cold”, K17 expression was inversely correlated with CD8+ cell infiltration, patients’ response to immune check point therapy, and in vivo tumor growth in mice [15]. The effect of K17 on immune regulation was at least partially dependent upon regulation of the CXCL9/CXCR3 axis in the tumor microenvironment [15]. On the other hand, another group found higher K17 expression in dMMR colon cancers compared to pMMR colon cancer [66]. Colon cancers bearing deficiencies in the mismatch repair machinery are considered immunologically »warm« and responded better to checkpoint blockade therapy than colon cancers that are mismatch repair proficient (pMMR) [93]. Using immunologically »warm« murine syngeneic colon cancer models, CT26 and MC38, they demonstrated that overexpression of K17 in colon cancer cells was positively correlated with CXCL10 expression by cancer cells, T cell infiltration, and response to check point blockade therapy [66]. K17 is known to promote CXCL10 and CXCL11 expression in skin tumor keratinocytes [94]. The recent data in colon cancer models [66] are consistent with the prior finding that K17 expression in tumor cells enhances tumor cell intrinsic CXCL10 expression. CXCL9, CXCL10, and CXCL11 are three soluble ligands for CXCR3 [95]. Macrophage-derived CXCL9 and CXL10 have been recognized as markers for an improved response to immunotherapy in melanoma patients, and are required for the efficacy of immunotherapy in murine syngeneic models of MC38 (colon cancer), AT-3, E0771 (breast cancer), and B16F10 (melanoma) models [96]. Interestingly, in this study, the group has identified macrophages as the major source of CXCL9 and CXCL10 in multiple murine tumors. When AT-3 tumors were injected into CXCL9/CXCL10 double-knockout hosts or hosts depleted of macrophages, the anti-tumor response of the checkpoint blockade was completely lost [96]. This finding is consistent with what was reported using the MOC2 murine head and neck cancer model [15] in that macrophage-derived CXCL9 is also important in attracting CXCR3+ T cells. However, how K17 expression in tumor cells leads to decreased CXCL9 production in immune cells has yet to be determined. The crosstalk between K17-expressing tumor cells and the immune microenvironment needs to be addressed in future studies. 

The above data suggest that there are two possible major sources of CXCR3 ligands in the tumor microenvironment: tumor cells and myeloid cells. Depending on the major source of CXCR3 ligands in the tumors, K17 may play opposite roles in the total production of these ligands in tumors due to its opposite regulation for intrinsic ligand production in tumor cells vs. extrinsic ligand production in immune cells. Taken together, these data demonstrate the multifaceted role of K17 in cancer immunity and the potential for clinical translation as both a biomarker and a drug target.

Generally, the discovery of promising carcinogenic pathways and biomarkers translates into clinical practice rather slowly [97]. This is, in part, due to the nature of cancer as a continuously evolving disease, arising and progressing as a consequence of several genetic alterations in key cellular processes [98], which hinders the diagnostic and prognostic capability of a single biomarker. Additionally, biomarker validation studies are usually retrospective and limited in size, with variable methodologies and assays used. That is indeed a trend we have observed in this scoping review. Two thirds of the included studies were retrospective in their design. Although the majority (32/39) of protein-based studies (prognostic and predictive (Appendix A)) used IHC as a K17 assessment method, the antibodies and scoring methodologies varied widely between the studies. Almost half of these studies did not specify how the prognostic marker cut-off was determined. In addition, only one study included interobserver variability assessment. Even well-established methods such as IHC for PD-L1 are open to observer interpretation, necessitating robust assay validation [99]. The heterogeneity of the K17 protein assay in the studies included here represents a major limitation of this review. Considering the promising evidence for both the predictive and prognostic value of K17 in several cancer types, we believe future studies should focus on (1) assay validation, as this may assist in streamlining clinical translation; and (2) using prospective cohorts or retrospective analyses of well-curated biospecimens linked to high-quality clinical outcome data [90]. 

The limitations of the present study are inherent in the choice of a scoping review methodology. First, our analysis is limited by the available literature on the selected topic. It is likely that additional cancer types were investigated or screened for the prognostic significance of K17 but were not published due to a lack of association. In addition, the heterogeneity of the reported results and case selection hindered our ability to standardize data extraction and the presentation of results. Due to variable study populations, non-standardized assays between different studies, and variably reported effect sizes (some did not report effect size, some reported risk ratio (RR), others hazard ratio HR), we were unable to analyze the data in a systematic fashion or perform a risk of bias analysis of the evidence. The provision of implications for practice in the form of clinical recommendations is a key feature of systematic reviews and is recommended in the reporting guidelines for systematic reviews, but it is not the goal of scoping reviews [100]. Our goal was to provide an overview of the evidence and identify knowledge gaps, as is intended in scoping reviews [32,100].

## 5. Conclusions

The prognostic and predictive significance of K17 varies by cancer type; however, in most cancer types, K17 was found to be a negative prognostic factor. Predictive studies in larger validation cohorts are lacking. Future work should focus on establishing a reproducible, clinically validated K17 assay.

## Figures and Tables

**Figure 1 viruses-15-02320-f001:**
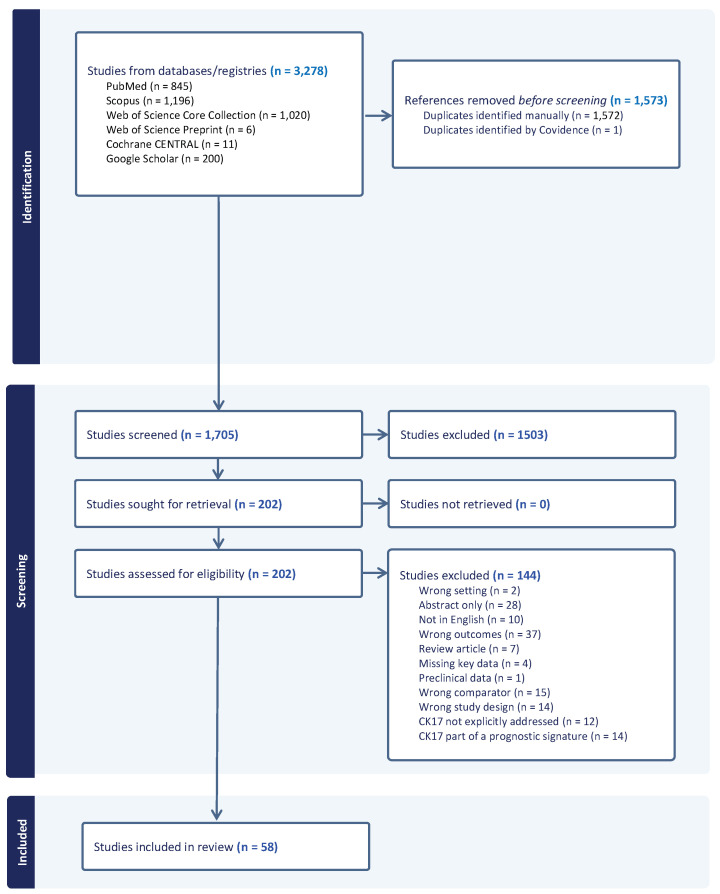
Flow chart of study selection and identification (PRISMA).

**Figure 2 viruses-15-02320-f002:**
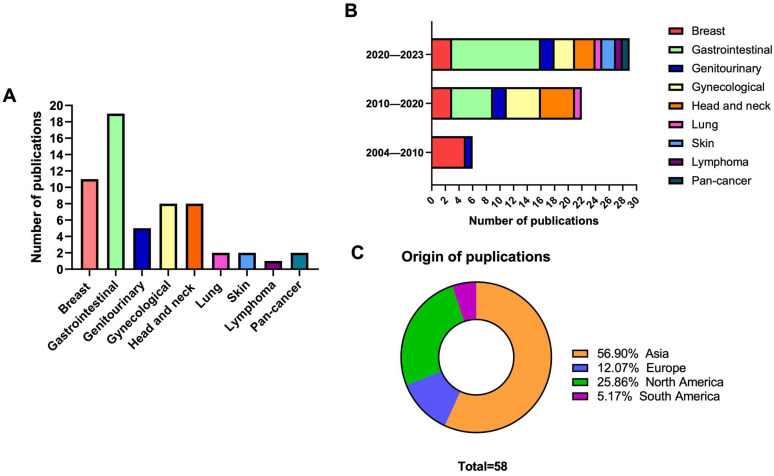
The characteristics of included studies: (**A**) number of publications per cancer type; (**B**) number of publications relative to year of publication; (**C**) origin of publications.

**Figure 3 viruses-15-02320-f003:**
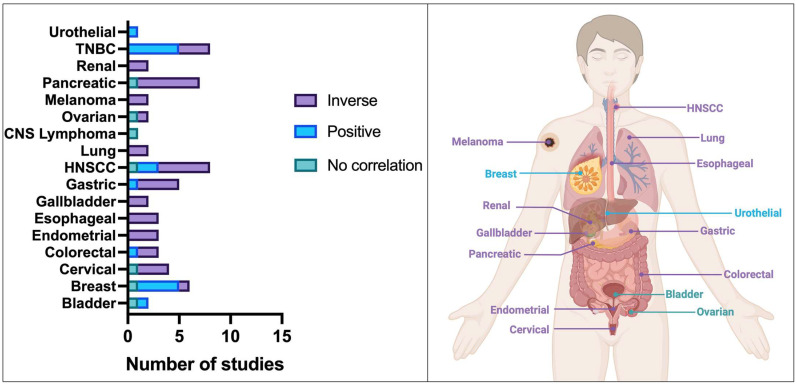
Summary of prognostic studies (pooled RNA and protein expression) by cancer type. The direction of the association between stress keratin 17 expression and outcome is color coded. Right panel represents the intepretation of reported findings per cancer type. The interpretation was based on >50% studies with concordant findings. Abbreviations: HNSCC—head and neck squamous cell carcinoma; TNBC—triple-negative breast cancer.

**Table 1 viruses-15-02320-t001:** Characteristics of included studies. Abbreviations: HNSCC—head and neck squamous cell carcinoma; SCC—squamous cell carcinoma.

	N (%)
**K17 evaluation method**	
Protein	33 (56.9%)
RNA	20 (34.5%)
RNA and Protein	5 (8.6%)
**Cancer type**	
Breast	11 (19.0%)
Gastrointestinal	19 (32.8%)
Genitourinary	5 (8.6%)
Gynecologic	8 (13.8%)
Head and Neck	8 (13.8%)
Lung	2 (3.4%)
Lymphoma	1 (1.7%)
Skin	2 (3.4%)
Pan-cancer	2 (3.4%)
**HPV-related cancers**	
Oropharyngeal HNSCC	2 (3.4%)
Non-oropharyngeal/any HNSCC	6 (10.3%)
Cervical SCC	2 (3.4%)
Cervical, any	1 (1.7%)
**Study design**	
Retrospective	44 (75.9%)
Prospective	8 (13.8%)
Online database/computational	6 (10.3%)
**Study objective**	
K17 only	48 (82.8%)
K17 in a gene signature	4 (6.8%)
K17 in a protein panel	6 (10.3%)
**Clinical significance**	
Prognostic	54 (93.1%)
Predictive	4 (6.9%)
immunotherapy	2 (3.4%)
chemotherapy	2 (3.4%)

**Table 2 viruses-15-02320-t002:** Summary of prognostic studies by cancer type. Number of studies indicates the number of studies that have investigated a given cancer type (pan-cancer studies excluded). Negative correlation confers a negative prognostic value, whereas positive correlation means a positive prognostic value.

Cancer Type	Evaluation Method	N (All Studies)	N (Negative Correlation)	N (Positive Correlation)	N (No Correlation)
Bladder	Protein	2	-	1	1
Breast	Protein	3	1	2	1 *
	RNA	2	-	2	-
Cervical	Protein	4	3	-	1
Colorectal	Protein	3	2	1	-
Endometrial	Protein	1	1	-	-
	RNA	2	2	-	-
Esophageal	Protein	2	2	-	-
	RNA	1	1	-	-
Gallbladder	Protein	2	2	-	-
Gastric	Protein	3	3	-	-
	RNA	2	1	1	-
HNSCC	Protein	5	2	2	1
	RNA	3	3	-	-
Lung	RNA	2	2	-	-
Lymphoma	RNA	1	-	-	1
Ovarian	Protein	2	1	-	1
Melanoma	RNA	2	2	-	-
Pancreatic	Protein	3	3	-	-
	RNA	4	3	-	1
Renal	Protein	1	1	-	-
	RNA	1	1	-	-
TNBC	Protein	7	2	5	-
	RNA	1	1	-	-
Urothelial **	Protein	1	-	1	-

* No correlation overall but found positive association in HER2+ and inverse in HER2 low breast cancer. ** Urothelial carcinoma of the upper urinary tract. Abbreviations: HNSCC—head and neck squamous cell carcinoma; TNBC—triple-negative breast cancer.

**Table 3 viruses-15-02320-t003:** Prognostic studies evaluating stress keratin 17 (K17) protein expression. Abbreviations: HNSCC—head and neck squamous cell carcinoma; SCC—squamous cell carcinoma; TNBC—triple-negative breast cancer; IHC—immunohistochemistry; IF—immunofluorescence; TMA—tissue microarray; R—retrospective; P—prospective; NR—not reported; OS—overall survival; DFS—disease-free survival; DSS—disease-specific survival; DCR—disease control rate; EFS—event-free survival; MFS—metastasis-free survival; RFS—relapse-free survival; ROC—receiver operating characteristic; HR—hazard ratio.

Author, Year (Ref)	Cancer Type	Evaluation Method	N	%K17 High Expressors	Endpoint	Correlation between K17 and Outcome	Effect Size
**Wu (2021)** [64]	Bladder	IHC; TMA	101	55%	OS	positive	Univariate HR (high): 0.11, *p* < 0.01 Multivariate HR (low): 4.263; *p* = 0.019
**Ingenwerth (2022)** [67]	Bladder	IHC	190	54%	DSS	no correlation	NR
**Rodriguez-Pinilla (2007)** [37]	Breast	IHC; TMA	245	12%	MFS, OS	negative	Univariate RR (high): 1.22, *p* = NS for MFS and HR = 1.8, *p* = NS for OS
**Tang (2022)** [65]	Breast	IHC; Transcriptome	150	NR	OS	positive in HER2high; inverse in HER2low	Univariate HR (low): 0.6, *p* = 0.002
**He (2021)** [34] *****	Cervical	ELISA	134	NR	OS	negative	NR
**Mockler (2017)** [23]	Cervical adenocarcinoma	IHC	90	<40%: 64% >90%: 13%	OS	negative	Univariate HR (high): 3.47, *p* = 0.013 Multivariate HR (high): 2.76, *p* = 0.048
**Escobar-Hoyos (2014)** [22]	Cervical SCC	IHC; TMA	65	35%	OS	negative	NR
**Hashiguchi (2019)** [53]	Cervical SCC	IHC	129	60%	OS	no correlation	Univariate HR (high): 0.56, *p* = NS Multivariate HR: 0.65, *p* = 0.3
**Ji (2021)** [54]	Colon adenocarcinoma	IHC	78	50%	Survival state	negative	NR
**Ujiie (2020)** [55]	Colorectal	IHC; Transcriptome; TMA	154 (Cohort 1 = 110, Cohort 2 = 44)	50%	RFS	negative	Univariate HR (high): 5.30, *p* < 0.001 Multivariate HR (high): 7.81, *p* < 0.001
**Liang (2023)** [66]	Colorectal	IHC	446	NR	OS, recurrence rate	positive	Univariate HR (high): 0.33; *p* = 0.0004 for OS and HR = 0.33, *p* < 0.01 for DFS
**Bai (2019)** [30]	Endometrial	IHC; Transcriptome	119	39%	OS	negative	Univariate HR (high): 1.75, *p* = 0.049 Multivariate HR (high): 2.0, *p* = 0.019
**Wang (2013)** [27]	Epithelial ovarian cancer	IHC	104	54%	OS	negative	Univariate RR (high): 1.44, *p* < 0.01 Multivariate RR (high): 1.531, *p* < 0.01
**Liu (2020)** [56]	Esophageal SCC	IHC; TMA	64	66%	OS	negative	Univariate HR (high): 2.51, *p* = 0.045 Multivariate HR (high): 5.383, *p* = 0.012
**Haye (2021)** [57]	Esophageal SCC	IHC; Transcriptome	68	58%	EFS	negative	Univariate HR (high, advanced stages subgroup): 2.08, *p* = 0.0384.
**Carrasco (2021)** [58]	Gallbladder	IHC; TMA	162	73%	OS	negative	Univariate HR (high, poorly differentiated): 2.0, *p* < 0.01, Multivariate HR (high, poorly differentiated): 2.46, *p* = 0.037
**Kim (2017)** [59]	Gallbladder adenocarcinoma	IHC; TMA	77	53%	DSS	negative	Univariate HR (high): 4.76, *p* = 0.001 Multivariate HR (high): 3.62, *p* = 0.01
**Alkhasawneh (2016)** [61]	Gastric	IHC	63	51%	OS	negative	NR
**Hu (2018)** [60]	Gastric	IHC; TMA	569	56%	OS	negative	Univariate HR (high): 1.454, *p* < 0.01 Multivariate HR (high): 1.336, *p* < 0.01
**Ide (2012)** [26]	Gastric adenocarcinoma	IHC; TMA	192	50%	5-year DSS	negative	Univariate HR (low): 0.51, *p* = 0.004 Multivariate HR (low): 0.786, *p* = 0.049
**Xu (2018)** [39]	HNSCC	IHC; TMA	106	54%	DSS	positive	Univariate HR (low): 2.04, *p* = 0.022 Multivariate HR (low): 1.854, *p* = NS
**Wang (2022)** [15]	HNSCC	IF	107	70%	OS, DCR	negative	NR
**Coelho (2015)** [41]	HNSCC, oral	IHC; TMA	67	79%	DFS, DSS	positive	Multivariate HR (low) in subgroup treated with surgery and RT: 4.11; *p* = 0.027 for DFS and 4.75; *p* = 0.016 for DSS
**Tojyo (2019)** [40]	HNSCC, oral	IHC	49	20%	5-year DFS	no correlation	NR
**Regenbogen (2018)** [24]	HNSCC	IHC	78	96%	survival state, OS	negative	Multivariate HR (high): 2.30, *p* = 0.0152
**Dundr (2022)** [68]	Ovarian	IHC; TMA	125	14%	OS, DFS, LFS, MFS	no correlation	NR
**Roa-Peña (2019)** [31]	Pancreatic	IHC; Transcriptome	74	24%	OS	negative	Univariate HR (high): 2.96, *p* = 0.008 Multivariate HR (high): 3.09, *p* = 0.015
**Roa-Peña (2021)** [35] ******	Pancreatic	IHC	211 ***	35%	OS	negative	Univariate HR (high, combined cohorts): 1.7, *p* = 0.0017, Multivariate HR (high, discovery set): 1.9, *p* = 0.0235
**Kawalerski (2022)** [48]	Pancreatic	IHC; Mass spec	26	23%	OS	negative	Univariate HR (highly detergent-soluble): 2.854; P = 0.046
**Sarlos (2019)** [62]	Renal cell	IHC; TMA	692	14%	DSS, recurrence	negative	RR (high) for post-operative recurrence: 2.5, *p* < 0.01
**Liu (2009)** [43]	TNBC	IHC	112	34%	RFS, OS	negative	Univariate HR (high): RFS: HR = 2.121, *p* = NS for RFS and HR = 2.142, *p*= 0.037 for OS Multivariate: HR = 0.910, *p* = NS for RFS and HR= 0.933, *p* = NS for OS
**Dogu (2010)** [46]	TNBC	IHC	33	74%	DFS	no correlation	NR
**Thike (2010)** [63]	TNBC	IHC; TMA	653	NR	DFS	negative	NR
**Cho (2011)** [45]	TNBC	IHC	88	53%	RFS < 36 months	no correlation	NR
**Kraus (2012)** [47]	TNBC	IHC; TMA	56	45%	PCR	no correlation	NR
**Merkin (2017)** [29]	TNBC	IHC; Transcriptome	149	>0%: 82%	EFS	no correlation	NR
**daSilva (2021)** [70]	TNBC	IHC; TMA	168	91%	EFS, OS	no correlation	NR
**Langner (2004)** [36]	Urothelial	IHC; TMA	53	40%	MFS	positive	NR

* used serum samples. ** used histological and cytological samples. *** discovery set: 106 and validation set: 105.

**Table 4 viruses-15-02320-t004:** Prognostic studies evaluating stress keratin 17 (K17) RNA expression. Abbreviations: HNSCC—head and neck squamous cell carcinoma; SCC—squamous cell carcinoma; TNBC—triple-negative breast cancer; NSCLC—non-small cell lung cancer; R—retrospective; P—prospective; NR—not reported; OS—overall survival; DFS—disease-free survival; EFS—event-free survival; RFS—relapse-free survival; ROC—receiver operating characteristic; HR—hazard ratio; LIHC—liver hepatocellular carcinoma; PAAD—pancreatic adenocarcinoma; UCEC—uterine corpus endometrial carcinoma; BC—breast invasive carcinoma; BLCA—bladder urothelial carcinoma; THCA—thyroid carcinoma; SKCM—skin cutaneous melanoma; ccRCC—clear-cell renal cell carcinoma; pRCC—papillary renal cell carcinoma.

Study ID (Ref)	Cancer Type	Data Derived from	Sample Size	K17 Cut-Off	Endpoint	Correlation between K17 and Outcome	Effect Size
**Tang (2022)** [65]	Breast	KM Plotter	NR	Not specified	OS	Positive	HR (high): 0.60, *p* = 0.002
**Modi (2022)** [78]	Breast	KM Plotter	1070	50%	OS	Positive	Univariate HR (high): 0.59, *p* = 0.0017
**Bai (2019)** [30]	Endometrial	TCGA	271	NR, determined experimentally, 39% K17 high	OS	negative	Univariate HR (high): 1.8, *p* = 0.01, Multivariate HR (high): 1.4, *p* = 0.086
**Zhang (2022)** [71]	Endometrial	TCGA	552	Median	OS	negative	Univariate HR (high): 1.65, *p* = 0.018, Multivariate HR (high): 1.29, *p* = 0.409
**Haye (2021)** [57]	Esophageal SCC	TCGA	90	NR, determined experimentally, 58% K17 high	EFS	negative	Univariate HR (high): 2.17, *p* = 0.04, Multivariate HR(high): 2.4, *p* = 0.04
**Zhou (2021)** [72]	Gastric	KM Plotter	635	NR, determined experimentally	OS	negative	Univariate HR (high, validation cohort): 1.59, *p* < 0.01
**Li (2022)** [82]	Gastric cancer	TCGA	diffuse: 135, intestinal: 146	50%	OS	Positive	NR
**Wang (2020)** [42]	HNSCC, oral	in study (qRT-PCR)	135	50%	OS	negative	Multivariate HR (high): 2.49 *p* = 0.004
**Wang (2022)** [52]	HNSCC, laryngeal	in study	42	Median	OS	negative	NR
**Kitamura (2017)** [38]	HNSCC, oral	in study (PBMC)	19	32%, determined experimentally	DFS	negative	NR
**Luo (2021)** [73]	Lung adenocarcinoma	TCGA	500	NR, determined experimentally	OS	negative	Multivariate HR (high): 1.1, *p* = 0.028
**Wang (2019)** [74]	Lung; NSCLC	UALCAN, KM Plotter	NR	Not specified	OS	negative (KM Plotter cohort only)	Univariate HR (high): 1.45, *p* = 1.1 × 10^−8^
**Han (2021)** [75]	Melanoma	TCGA, GEO	458	50%	OS	negative	Univariate HR (high): 1.5, *p* = 0.0018
**Miñoza (2022)** [76]	Melanoma	TCGA	468	50%	OS	negative	NR
**Li (2021)** [80]	Pan-cancer	TCGA, GEO	NR	50%	OS, DFS	negative	For OS: SKCM: n = 515; HR (high, OS) 1.4, *p* = 0.038; PAAD: 364; HR 1.5; *p* = 0.017; mesothelioma 82; HR 3.1, *p* = 2.3 × 10^−5^; LUAD: 178, HR 1.5, *p* = 0.038; LIHC: 478, HR 1.6, *p* = 0.0033; RCC: 458, HR 1.5; *p* = 0.0018. For DFS: mesothelioma: 82, HR (high) 1.8; *p* = 0.048; PAAD: 178, HR 1.7, *p* = 0.017
**Zhang (2022)** [71]	Pan-cancer	KM Plotter	NR	Median	OS, RFS	negative and positive	For OS: RCC: HR (high) 1.64, *p* = 0.003, LIHC: HR 1.63, *p* = 0.005, LUAD: HR 1.53, *p* = 0.007, PAAD: HR 1.99, *p* = 0.008, UCEC: HR 1.89, *p* = 0.004, BC: HR (high) 0.55, *p* < 0.001. For RFS: THCA: HR 0.41, *p* = 0.04, RCC: HR 0.15, *p* = 0.034, UCEC: HR 0.5, *p* = 0.008, BLCA: HR 2.19, *p* = 0.026, PAAD: HR 2.61, *p* = 0.018
**Roa-Peña (2019)** [31]	Pancreatic	TCGA, prior literature	Cohort 1: 145, Cohort 2: 124	76%, determined experimentally	OS	negative	Univariate HR (high, combined): 1.69, *p* = 0.003 Multivariate HR (high): 1.79, *p* = 0.037
**Li (2022)** [49]	Pancreatic	TCGA	178	Median	OS	No correlation	NR
**Lu (2021)** [50]	Pancreatic adenocarcinoma	TCGA	170	Median TPM	OS	negative	Univariate HR (high): 1.15, *p* = 0.018
**Stone (2018)** [51]	Pancreatic adenocarcinoma	in study	24	Median	OS	negative	NR
**Takashima (2021)** [79]	Primary CNS lymphoma	in study	31	50%	OS	No correlation	NR
**Wach (2019)** [77]	Renal	TCGA	ccRCC n = 522; pRCC: 284	Median	OS	negative	NR
**Merkin (2017)** [29]	TNBC	TCGA	149 (IDC only)	NR, determined experimentally	EFS	negative (subgroup only)	NR

**Table 5 viruses-15-02320-t005:** Studies evaluating the predictive role of stress keratin 17 (K17) in human cancers. Abbreviations: HNSCC—head and neck squamous cell carcinoma; R—retrospective; P—prospective; NR—not reported; IHC—immunohistochemistry; TMA—tissue microarray; dMMR—miss match repair deficient; OS—overall survival; PFS—progression-free survival; DCR—disease control rate; HR—hazard ratio.

Study ID	Cancer Type	Study Type	Study Design	K17 Evaluation Method	Treatment	Sample Size	% K17 High	Endpoint	Correlation between K17 and Outcome	Effect Size
**Diallo (2006)** [69]	Breast	Protein	P	IHC; TMA	High-dose (HDCT) vs. Dose dense chemotherapy (DDCT)	224	6%	OS	negative	Univariate HR (high, DDCT arm): 5.1, *p* < 0.001
**Liang (2023)** [66]	Colorectal	Protein	R	IHC	anti PD-1 in dMMR patients	30	NR	objective response to therapy (RECIST)	positive	NR
**Wang (2022)** [15]	HNSCC	Protein	R	IHC; TMA	Pembrolizumab	26	70%	DCR, PFS, OS	negative	NR
**Pan (2020)** [83]	Pancreatic	RNA	R	Transcriptome (APGI)	Gemcitabine	94	24%	OS	negative	Univariate HR (high,): 1.8, *p* = 0.335, Multivariate HR: 1.79, *p* = 0.046

## Data Availability

All data generated as part of this publication has been made available.

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
