# Peer review of "Emerging Prognostic and Predictive Significance of Stress Keratin 17 in HPV-Associated and Non HPV-Associated Human Cancers: A Scoping Review"

_viruses, 2023, doi:10.3390/v15122320_

Round 1

Reviewer 1 Report

Comments and Suggestions for Authors

The manuscript, entitled, "Emerging prognostic and predictive significance of stress 2 keratin 17 in viral and non viral human cancers: a scoping review" was prepared using the Preferred Reporting Items for Systematic Re-90 views Extension for Scoping Reviews (PRISMA-ScR) guidelines. A scoping review should identify knowledge gaps, examine the research methods commonly used, and determine if a systematic review would be useful. Here, the authors asked, What is the prognostic and predictive significance of stress keratin 17 expression in human cancer patients?  An extensive database search was performed, using specific key words, with the help for a research librarian, to identify publications that should be included based on the inclusion criteria. These were: publications were published in English, involved human studies, directly evaluated K17 status at the RNA or protein level  and addressed the association between K17 status and clinical outcome. Studies that only evaluated K17 as part of a panel were listed in supplemental tables.  The authors determined, as expected, that the prognostic and predictive value of K17 varied by cancer site, although many studies showed an inverse correlation between expression and outcomes.

However, the authors point out that there were many differences in assays, tissue staining evaluation, and tumor types in the evaluate studies that hinder the use of using a single biomarker for diagnostic or prognostic evaluation. In addition, studies on tumor subtypes show varying results. Impact of K17 on multiple systems, including the tumor microenvironment and the immune systems, complicate developing a clear picture of how K17 expression should be used in the clinic, but this scoping review will be a good source of compiled information on those that are interested in advancing this field of knowledge, perhaps performing a systematic review with a specific tumor type/treatment model in mind.

Author Response

We would like to thank the reviewer for taking the time to review this manuscript. We appreciate your positive critique.

Reviewer 2 Report

Comments and Suggestions for Authors

The review "Emerging prognostic and predictive significance of stress keratin 17 in viral and non viral human cancers: a scoping review" shows an growing interest in the expression of cytokeratin 17 (K17) and correlates with inferior clinical outcomes across various cancer types. The author suggest that K17 is a negative prognostic factor in the majority of studied cancers and in three out of four predictive studies, K17 was a negative predictive factor for chemotherapy and immune checkpoint blockade therapy response. Only 2/17 studied cancer types showed a positive prognostic factor. 

Major issues with this manuscript:

The authors need to talk more about viral and non-viral cancers in this manuscript since it is in the headline and the author stated in the introduction, that in this scoping review they present human cancers, particularly those of viral origin.The author only mentioned briefly mouse papillomavirus and is missing to talk about human viral cancers.

Most of the cancers discussed in this review are not of viral origin. This needs to be revised.

All the tables need to be better organized. For example: A bold headline would help for better understanding.

Minor issues:

The references need to be checked. It is not consistent. For example, one time the references are ending before the period, then after the period. Sometimes there are gaps sometimes not. In line 269 is a different font for the references. 

Author Response

We would like to thank the reviewer for taking the time to review this manuscript and for their helpful suggestions on how to improve this manuscript. We have addressed your suggestions as follows:

- Unfortunately, most of the studies on the clinical significance of K17 in human cancers were performed in cancers of non-viral origin. We do, however, believe the present manuscript addresses an important knowledge gap on how research into the clinical relevance of K17 is currently conducted and which cancer types may be of interest for future studies. Our findings suggest head and neck cancer (a cancer type with a known association to HPV infection relative to anatomic location) is among the cancer types with the strongest evidence for clinical translation. This commentary has been added to the Discussion section, p. 15, lines 335-341. We discuss the data on the role of K17 in HPV-associated cancers in the Results section, p. 9, lines 234-239 as well as in the Discussion section, p. 16, lines 394-410. Given the preclinical observations that the presence of papillomavirus can induce the expression of K17, we have restructured the paper to specifically address the correlation with HPV infection. The title has been changed to '' Emerging prognostic and predictive significance of stress keratin 17 in HPV-associated and non HPV-associated human cancers: a scoping review''. We've added a section on preclinical data suggesting K17 may be upregulated as a result of papillomavirus infection and it's effects on the host immune response (Introduction, page 2, line 75-89). Consistent with this change of narrative, changes have been made to the results section: page 5, lines 182-185; updated Table 1 to include number of studies in HPV-associated cancers, a paragraph on studies performed in HNSCC and cervical cancer on page 6, lines 217-235; and updated the discussion section lines 368-375. We have also made minor changes to the abstract.

- We've made minor changes to the tables to improve clarity as suggested by reviewer. The first rows and columns of each table have been changed to bold headlines. We believe the current format of the tables follows the standard format used in scoping and systematic reviews. Significant efforts have been put into condensing the tables to include only the key findings per study, and detailed supplemental tables have been made available to the readers.

- The inconsistencies in referencing style have been fixed and all references are now ending before the period.

Round 2

Reviewer 2 Report

Comments and Suggestions for Authors

The authors addressed all comments and I do not have any more suggestions.